# The Beneficial Role of Photobiomodulation in Neurodegenerative Diseases

**DOI:** 10.3390/biomedicines11071828

**Published:** 2023-06-26

**Authors:** Ayodeji Abijo, Chun-Yuan Lee, Chien-Ying Huang, Pei-Chuan Ho, Kuen-Jer Tsai

**Affiliations:** 1Taiwan International Graduate Program in Interdisciplinary Neuroscience, National Cheng Kung University and Academia Sinica, Taipei 11529, Taiwan; abijoayodeji@gmail.com; 2Institute of Clinical Medicine, College of Medicine, National Cheng Kung University, Tainan 70101, Taiwan; peggy821124@gmail.com; 3Neurobiology Unit, Department of Anatomy, Ben S. Carson School of Medicine, Babcock University, Ilishan-Remo 121003, Nigeria; 4Aether Services, Taiwan, Ltd., Hsinchu 30078, Taiwan; 5S-Radiant LED Technology Co., Ltd., Hsinchu 30042, Taiwan; jenyin.huang@s-radiantled.com.tw; 6Center of Clinical Medicine, National Cheng Kung University Hospital, College of Medicine, National Cheng Kung University, Tainan 70101, Taiwan

**Keywords:** photobiomodulation, low laser light therapy, light-emitting diode (LED), neurodegenerative diseases, therapy

## Abstract

Photobiomodulation (PBM), also known as Low-level Laser Therapy (LLLT), involves the use of light from a laser or light-emitting diode (LED) in the treatment of various disorders and it has recently gained increasing interest. Progressive neuronal loss with attendant consequences such as cognitive and/or motor decline characterize neurodegenerative diseases. The available therapeutic drugs have only been able to provide symptomatic relief and may also present with some side effects, thus precluding their use in treatment. Recently, there has been an exponential increase in interest and attention in the use of PBM as a therapy in various neurodegenerative diseases in animal studies. Because of the financial and social burden of neurodegenerative diseases on the sufferers and the need for the discovery of potential therapeutic inventions in their management, it is pertinent to examine the beneficial effects of PBM and the various cellular mechanisms by which it modulates neural activity. Here, we highlight the various ways by which PBM may possess beneficial effects on neural activity and has been reported in various neurodegenerative conditions (Alzheimer’s disease, Parkinson’s disease, epilepsy, TBI, stroke) with the hope that it may serve as an alternative therapy in the management of neurodegenerative diseases because of the biological side effects associated with drugs currently used in the treatment of neurodegenerative diseases.

## 1. Introduction

Low-level laser therapy (LLLT) also known as photobiomodulation (PBM) is currently being explored because of its known therapeutic potential. The idea that the energy of incident light excites electrons in chromophores and causes them to hop from lower-energy orbits to higher-energy orbits is the foundation of the scientific discipline of photobiology.

Low-level laser treatment or photobiomodulation involves the application of light for therapeutic purposes. Therapeutically, light in the red to near infrared spectrum is used in this practice (Figure 1). The delivered fluence is not known to cause any ablative effect. An underlying photochemical mechanism provides support for the PBM. The most common wavelength used for this purpose is between 650 and 1200 nm, which is absorbed by cellular chromophores [1]. Mitochondrial cytochrome c oxidase (CCO), electron chain transport complex IV, absorbs light with a wavelength range of 600 to 900 nm. It is believed that longer wavelengths of light activate water- and light-sensitive ion channels such as those in the transient receptor potential (TRP) family [1].

Historically, Endre Mester of Hungary accidentally discovered photobiomodulation (PBM) in 1967 when he tried to replicate an experiment published by McGuff of Boston, United States [2]. A tumor that had been surgically implanted into a cancerous lab rat was destroyed by a beam from the newly discovered ruby laser [3]. used by McGuff. The ruby laser made for Mester was not nearly as powerful as the laser McGuff had previously used. Instead of curing the experimental tumors with his low-powered laser, Mester was able to induce hair regrowth and wound healing in the rodents at the sites where the tumors were implanted [1,4,5].

Because of this finding, several articles were published detailing what Mester called “laser biostimulation” (LLLT). The low optical fluence prevents thermal and abrasive consequences, hence making it a safe procedure. Specifically, the PBM utilizes a photochemical process. Light with a frequency between 650 and 1200 nm is typically used for the procedure, as this range is optimal for absorption by cellular chromophores [6]. According to the first law of photobiology, for any photochemical reaction to take place within a biological tissue, the absorption bands of the chromophores present within the tissue must be able to absorb photons [7,8].

The role PBM plays in the mitochondria has brought it to the forefront as a potential therapeutic candidate in the treatment of neurodegenerative diseases, which are currently incurable due to their complex and multifaceted nature [1]. Studies have also demonstrated the neuroprotective efficacy of PBM in several models of neurodegenerative diseases [6]. However, some of the mechanisms by which PBM acts at the cellular level are still the subject of investigation.

Multiple pathways; ERK [9], Akt/GSK3β/β-catenin [10], Src/Syk/PI3K/Akt [11], and cAMP/PKA/SIRT1 [12] have been reported to be activated following biostimulation, as shown in Figure 2. This review, therefore, examines the role of PBM in several neurodegenerative diseases with the hope that it may serve as an alternative therapy in certain drug-refractory neurodegenerative diseases and, by extension, drugs with biological side effects.

## 2. Historical Perspective of Light Therapy

Historically, the concept of light therapy began with the use of sunlight (heliotherapy) and ultraviolet light (UV) in the treatment of various conditions and which has recently been referred to as photobiomodulation. The use of light for therapeutic purposes dates back to the 1850s when Florence Nightingale advocated for the use of sunlight and clean air in the management of several medical conditions [13]., with its roots in ancient Egypt, Greece, and India through to the medieval period. Born in 1860, Neils Finsen researched the use of light, perhaps motivated by the condition from which he suffered, Niemann–Pick disease, which reportedly, he had to sunbathe frequently to be able to examine the healing properties of light in living tissues. As a result of this, in 1880 Finsen reported a cure rate of 98% in 2000 patients with skin diseases and mucous membrane disorders treated with ultraviolet light. Dr. Neils Finsen was awarded a Nobel Prize in the early twentieth century (1903) and was also awarded the prize for Therapeutics by the University of Edinburgh (1904) as a result of the significant breakthrough made in the use of phototherapy (using ultraviolet light and red light) in the management of tuberculosis and smallpox, respectively [14]. In 1904, Finsen suffered from ascites and a heart condition and died at the age of 44. These significant discoveries by Finsen perhaps opened several doors in the field of phototherapy.

Surgeons required a dependable instrument that functioned to control postoperative infections before the development of antibiotics and sulfa medicines in the early 1940s. Emmett K. Knott (1928) was the first to irradiate human blood with ultraviolet light; the patient he treated had been abandoned after a miscarriage due to septicemia caused by hemolytic streptococcus. The patient responded amazingly to light irradiation and recovered in just two days. In 1942, Hancock and Knott treated 6520 individuals, all of whom showed no ill effects from their therapy for acute ear infections. In 1945, Levinson proved that ultraviolet light could be used to render viruses inert so that they might be used as vaccine antigens [15].

Since the 1950s, the use of ultraviolet light by dermatologists and physiotherapists in the management of several skin conditions has gained wide acceptability. Furthermore, in the 1980s and 1990s, there was an upsurge in the use of light therapy in various disciplines [16]. Without the successes documented in the use of light therapy over the years, our attention would not have focused on this important concept. Even though attention was shifted away from this concept for a few decades, the increasing amount of research into the application of light which is now referred to as photobiomodulation in the management of several conditions, most importantly neurodegenerative diseases, has re-ignited interest in this field. As a non-pharmacological approach with no perceived side effects, it is germanely something to consider.

## 3. Photobiomodulation Mechanisms

The increasing pieces of evidence on the use of photobiomodulation are based on the premise that the present therapeutic strategies that are targeted toward the treatment of most neurodegenerative diseases facilitate only symptomatic relief directed at the pathological hallmarks rather than slowing down, reversing, and/or alleviating disease progression. Some of the proposed mechanisms by which photobiomodulation, which is gaining increasing attention in biomedicine, exerts beneficial effects in certain neurodegenerative diseases are discussed below.

## 4. Role of Gated Channels in Photobiomodulation

Initially, there was limited evidence suggesting that light-gated or heat-gated ion channels are mechanisms of action in PBM, but this is beginning to change [17]. Transient receptor potential (TRP) channels may be impacted by PBM.

The TRPV “vanilloid” subfamily of TRP channels has been the primary focus of studies connecting PBM to light/heat-gated ion channels. Yang et al. [18], Ryu et al. [19], Albert et al. [20], and Gu et al. [21] had earlier hypothesized that perhaps the activation of TRP channels could be by green and infrared light. However, green light is not useful in clinical settings because it cannot penetrate tissue as effectively as infrared or near-infrared light can. Infrared (2780 nm) wavelength light was found by Ryu et al. [19] to suppress TRPV1 activation, resulting in less pain stimulus production. TRPV4 was also found to have an anti-nociceptive effect when exposed to light of the same wavelength, albeit one that was much less pronounced. In addition, Albert et al. also found that 1875 nm pulsed light activated TRPV4 [20].

## 5. ATP and Mitochondrial Function in PBM

ATP has long been recognized as the energy currency of the cell. However, recent evidence has also supported its role as a signaling molecule by producing certain cellular effects established through the activation of P2X and P2Y [22]. The report by Anders et al. [23]. demonstrated that normal human neural progenitor cells exposed to 810 nm of radiation expressed the P2Y2 and P2Y11 receptors in vitro. Concerning mitochondrial mechanisms of photobiomodulation, the many functions of ATP were addressed by Karu [22]. It has been known for decades that varying wavelengths of monochromatic light can stimulate the extra synthesis of ATP in isolated mitochondria and intact cells of different types [24,25]. This has also been long considered one of the most important mechanisms involved in photobiomodulation. In the mitochondrial respiratory chain cytochrome c oxidase (unit IV) is a key chromophore because it includes both heme and copper centers, allowing it to absorb light in the near-infrared spectrum. One popular theory holds that the photons release the enzyme’s inhibitory nitric oxide, thereby boosting electron transport, mitochondrial membrane potential, and ATP generation [26]. This provides supporting evidence that the mitochondria are the resident organelles for ATP production and oxidative phosphorylation and the enhancement of ATP following photobiostimulation.

## 6. Reactive Oxygen Species in PBM

Neurodegenerative illnesses and other neurological conditions are associated with elevated levels of reactive oxygen species (ROS), which can be mitigated through oxidative stress management [27,28] Inducible nitric oxide synthase regulation by cytokines is one such method (iNOS). Reducing reactive nitrogen species and oxidative stress is achievable when nitric oxide levels are lowered by inhibiting iNOS through antioxidant defense triggered by photobiomodulation.

It is conceivable that photobiostimulation can mitigate the effect of reactive oxygen species (ROS); these effects are the opposite of photodynamic therapy (PDT) which elevates reactive oxygen species. For therapeutic purposes and the stimulation of various cellular mechanisms, PBM utilizes non-ionizing radiation or light, particularly in the visible and near-infrared regions of the electromagnetic spectrum [29]. PBM causes free radical displacement and the body’s oxidative stress burden decreased when there is an antioxidant deficiency [1]. An increase in mitochondrial membrane potential (MMP), ATP, cyclic adenosine monophosphate (cAMP), and nitric oxide (NO) is necessary for PBM to exert its effects [26,30]. This is because cytochrome c oxidase is activated in this process thereby catalyzing the reduction of oxygen to water. The activation of specific cellular networks requires NO as a signaling agent. PBM boosts NO production by cleaving metal compounds in cytochrome c and/or increasing cytochrome c levels [31]. It is thought that PBM stimulates mitochondria-mediated cellular processes in damaged or dysfunctional tissue, leading to a broad variety of therapeutic effects [32]. While mitochondrial ROS production was apparent at longer wavelengths, some research suggests that light with a wavelength of 400–500 nm may generate free radicals through photosensitization. Gene transcription is triggered and reactive stress is mitigated due to PBM’s activation of cytosolic transcription factors such as NF-kB [33].

## 7. Light Properties as Consideration in Photobiomodulation

The electromagnetic spectrum comprises electromagnetic radiation (X-rays, gamma rays, ultraviolet rays, infrared, near-infrared, microwave, and radio waves) (Figure 1), whether visible or invisible to the human eye. Visible light is the type of light that can be seen by the naked human eye, while invisible light cannot be seen. Visible light spectrum oscillates between 400 and 700 nm wavelengths with near-infrared spectrum oscillating between 750 nm and 1400 nm. Lightwave propagation is somewhat similar to the ocean wave, however, the propagation of light waves involves alternating electromagnetic fields. Two important properties of light must be considered in the concept of photobiomodulation; these involve wavelength and frequency. In the electromagnetic spectrum, these two light properties are inversely related. This means that with increasing light frequency there is decreasing wavelength and vice-versa [13].

### 7.1. Wavelength

As discussed in the introductory aspect of this review, one of the important light properties to consider during the application of photobiomodulation is the wavelength. Light with a lower wavelength emits higher energy and vice-versa. In the visible light spectrum, light of a lower wavelength (blue light) emits higher energy and has deleterious effects on the retina, whereas light with a higher wavelength (red light) has lower energy [34]. In the visible light spectrum, the mnemonic, ROYGBV, which is used to represent red, orange, yellow, green, blue, and violet, represents the different colors that can be seen (Figure 1). Each of these colors has a different wavelength, and each falls within the lower, middle, and higher wavelength range [13] Blue light has a lower wavelength and emits higher energy while medium wavelength (yellow and green lights) emits middle energy and red light which has a higher wavelength emits a lower energy. The nomenclature “Low-Laser Light Therapy (LLLT)” ascribed to photobiomodulation probably stems from the low energy emitted by red to near-infrared (NIR) light. It is noteworthy that violet-blue light is hazardous just like blue light, while blue to green light with an approximate 480 nm wavelength has some roles to play in the pupillary light reflex and the circadian clock [35]. Jagdeo et al. [36] had also earlier reported the beneficial effects (anti-inflammatory and anti-bacteria) of the treatment of different skin conditions (psoriasis, acne, etc.) with blue light. However, because of the low-penetrating ability of blue light, its use is somewhat limited depending on the depth of the skin injury [37,38]. As opposed to blue light, red light requires a higher energy delivery to be able to deliver the same photons as blue light because of its higher wavelength; however, it has a greater skin penetrating ability as compared to blue light and its use has recently been approved by the FDA [13].

### 7.2. Photons (Energy)

Other important light characteristics such as the energy or photons are another important consideration in the field of photobiomodulation. German physicist, Max Planck, in the early 1900s, explained why light intensity decreases with increasing energy by postulating that it came from atoms vibrating at a particular frequency (v). He found that different particles might oscillate at various rates and according to Planck’s Law, the energy of an oscillation is proportional to its frequency. This formula E = *h*v was derived by Planck where E = Energy, *h* = Planck’s constant, and v = frequency and is now written as E = *h*c/λ, where λ is the wavelength and c is the velocity of light. This shows the relationship between photons and wavelength. These properties are to be considered by scientists in the field of phototherapy.

In photobiomodulation, light in the red to near-infrared spectrum is utilized in this practice. The delivered fluence is not known to cause any ablative effect and the energy emitted depends on the wavelengths within the red to near-infrared spectra [13]. An underlying photochemical reaction following stimulation provides support for the PBM.

### 7.3. Interaction between Light and Tissue in Photobiomodulation

According to Keiser [39], the interaction between light and tissue is indeed multifactorial. These factors may be summarized as reflection, refraction, absorption, and scattering of light photons (energy). Because of the structural heterogeneity of biological tissues and their different optical properties, these factors may influence interactions between light and biological tissues. Light being a wave can either be reflected through a media or refracted depending on the media or biological tissue it interacts with. The penetrating ability of light waves through a media (absorption) and the multiple scattering of photons as they pass through biological tissues together with the parameters of the light are important considerations when applying photobiomodulation. One would imagine that there would be differences between light penetrating biological tissues (the brain and bone) based on the structural differences. This implies that for different tissues, the light absorption rate differs. Absorbed light can either be converted, utilized in photochemical reactions, or applied in fluorescent microscopy. Light absorption is largely dependent on the wavelength.

Light scattering is a process that involves the diversion of light from the intended path to another path after absorption in biological tissues. Light scattering has also been reported to involve two mechanisms; whether elastic or inelastic, the former involves an unchanging wavelength with sustained photon energy with only a directional change during the scattering process, and the latter involves both directional change and change in wavelength (photon energy) during the scattering process [39].

## 8. Photobiostimulation in Neurodegenerative Diseases

### 8.1. Alzheimer’s Disease (AD)

Alzheimer’s disease is a progressive neurodegenerative condition and the most common form of dementia clinically characterized by cognitive and memory dysfunction [40]. and is associated with tau hyperphosphorylation and beta-amyloid protein aggregation. The treatments that are currently available for patients who suffer from Alzheimer’s disease are not very effective and come with constraints. In reality, these medications are ineffective for the vast majority of patients and are linked to a wide variety of adverse effects [41,42].

In the study by Michalikova et al. [43], a pretreatment with a daily 6 min exposure to near-infrared 1072 for 10 days resulted in significant behavioral effects in middle-aged female CD-1 mice (12 months) tested in a 3D-maze. Near-infrared treatment in middle-aged mice reversed the significant working memory deficits.

Transcranial Laser Therapy (TLT) has been hypothesized to maintain neuronal function by boosting ATP production, mitochondrial activity, and cellular respiration. To this end, a study was conducted to investigate how TLT influenced a transgenic mouse model of amyloid-protein precursor. TLT (808 nm, 0.5 W/cm^2^, 2.8 W/cm^2^ and 5.6 W/cm^2^, 675 J/cm^2^, 336 J/cm^2^, and 672 J/cm^2^ for 6 months) was administered three times weekly, beginning at the age of three months, with varying doses compared to a control group (no laser treatment) with treatments lasting 6 months. Amyloid load, inflammatory markers, brain and plasma beta-amyloid levels, cerebrospinal fluid beta-amyloid level, soluble amyloid precursor protein (sAPP) levels, and behavioral changes were assessed in the rats following the experimental period. The amyloid plaque accumulation in the brain was significantly decreased following TLT administration, and this effect was dose-dependent. Based on these findings, TLT showed promise in AD treatment [44].

Likewise, a narrow waveband of infrared light of 1072 nm that was administered on female TASTPM mice (2 months) for 6 min sessions for two consecutive days, biweekly for 5 months, showed significant results in decreasing amyloid plaques and upregulating heat-shock protein expression [45].

Purushothuman et al. [46] had also earlier reported that in two mouse models of AD (the K3 mice and APP/PS1 mice), photobiostimulation caused a significant reduction in Alzheimer’s disease markers and oxidative stress markers in both the cortex, hippocampus, and cerebellum [47].

The study by da Luz Eltchechem et al. [48] examined the impact of photobiomodulation (LED 627 nm) treatment for 21 days on spatial memory and behavioral state in rats with Alzheimer’s disease (AD) induced by intracerebral injection of Aβ25–35 toxins in the hippocampus. In their study, the PBM was placed in the frontal region of the rat and the irradiation was performed daily for 100 s at a dose of 7 J/cm^2^ with a potency of 70 mW. The result obtained from their study demonstrated that there was a significant reduction in beta-amyloid plaques as evaluated histologically following PBM treatment on days 7, 14, and 21. Spatial learning and memory assessment by the Morris water maze task and open field test conducted to assess several behaviors (locomotion, exploration, and anxiety) showed significant improvements following the 21 days.

An in vitro study by Sommer et al. [49] had earlier reported a reduction in beta-amyloid plaques following their incorporation into the human neuroblastoma cells (SH-EP cells) and treatment with light irradiation (600 nm).

Heo et al. [50] used a hippocampal cell line (HT-22) and mouse organotypic hippocampal tissues to examine the role of antioxidants, the BDNF expression, and antioxidant enzymes, as well as the activation of cAMP response element binding (CREB) and extracellular signal-regulated kinase (ERK) signal transduction pathways, to evaluate the effect of PBM on hippocampal oxidative stress. Photobiomodulation induced an increase in BDNF production by activating the ERK and CREB signaling pathways, protecting HT-22 cells from apoptosis. Additionally, the levels of phosphorylated ERK and CREB, which had dropped due to oxidative stress, and the expression of the antioxidant enzyme superoxide dismutase were all boosted by PBMT in hippocampal organotypic slices. This further substantiates the antioxidant role of photobiomodulation in counteracting oxidative stress and the likely beneficial role of photobiostimulation on hippocampal-dependent functions.

A study by Meng et al. [9] supported the beneficial role of photobiostimulation with evidence that PBM stimulates synaptogenesis and restoration of damaged synapses in chronic neurodegenerative disease, and acute or chronic traumatic brain injury occurs via upregulation of brain-derived neurotrophic factor (BDNF). In their study, they utilized the SH-SY5Y cell line and primary hippocampal neuronal cultures to be able to ascertain the effect of low-laser light treatment (LLLT) on beta amyloid-induced neuronal death and dendritic atrophy. They found that LLLT was able to attenuate beta amyloid-induced toxicity and neurodegeneration. Upregulation of BDNF was, however, ERK/CREB pathway-dependent which has also been documented earlier [51,52,53].

The role of glycogen synthase kinase beta (GSK-3β) in the hyperphosphorylation of the microtubule-associated protein (tau) and production of neurofibrillary tangles, one of the pathological hallmarks of AD, cannot be overemphasized [10]. Glycogen synthase 3 beta (GSK-3β) has also been known to have both apoptotic and anti-apoptotic functions [54,55,56]. In the same manner, the central nervous system development is dependent on Wnt/β-Catenin signaling, while β-Catenin can also be phosphorylated by GSK3β [10,57]. These pieces of evidence have demonstrated the roles of these pathways in the development and likewise as a potential target in the treatment of AD. To this end, the study by Liang et al. [10] found that low-power laser irradiation was able to inhibit the activity of GSK3β vis-à-vis activation of Akt which in turn inhibited beta amyloid-induced programmed cell death. This has elucidated the role that photobiomodulation has in the Akt/GSK3β/β-catenin pathway.

Inflammation, which is the response of living vascularized tissues to injury, is a major component of most neurodegenerative diseases. Glial cells (astrocytes and microglia) have been known to be activated during the inflammatory process in neurodegenerative diseases and also in the release of inflammatory molecules. Song et al. [11] sought to examine the role of LLLT on microgliosis and the downstream signaling events. In their study, microglial activation was modeled by lipopolysaccharide (LPS) treatment on microglia BV2 cells. They found that 20 J/cm^2^ LLLT diminished Toll-Like Receptors’ (TLR) inflammatory actions, evidenced by reduced expressions of cytokine and NO. It was concluded that Src/Syk/PI3K/Akt is involved in this perceived effect.

Sirtuins are a group of NAD+-dependent protein deacetylases that help fight age-related diseases such as cancer, diabetes, heart disease, and neurological diseases. The SIR2 ortholog Sirtuin1 (SIRT1) is one of the seven mammalian sirtuins and plays a crucial role in reducing the severity of neurodegenerative illnesses by modulating neuron survival, neurite outgrowth, synaptic plasticity, cognitive function, and neurogenesis [58,59]. SIRT1 has been reported to have neuroprotective functions in Alzheimer’s disease [59,60]. Because of the high expression of SIRT1 in the hippocampus, an important area for memory consolidation and also vulnerable to AD pathology, Zhang et al. [12] researched the role of photobiomodulation therapy in AD. They reported that photobiomodulation offered protection in AD pathology by facilitating a shift in the amyloid precursor protein to the non-amyloidogenic pathway through the activation of the AMP/PKA/SIRT1 pathway.

### 8.2. Parkinson’s Disease (PD)

After Alzheimer’s disease, the most common form of neurodegeneration is the movement disorder known as Parkinson’s disease (PD) [61]. About 7,000,000 people around the world suffer from this illness [62]. There are estimates that 4% of people over the age of 80 will develop this age-related condition. Currently, 1–2% of people over the age of 60 suffer from it [63]. Over 17,000 people are diagnosed with Parkinson’s disease each year [64]. It is more common in men and can appear as early as age 20 [65,66]. The primary cause of this disorder is the dopaminergic neuronal loss in the substantia nigra’s striatal-projecting pars compacta (SNc) [67]. Intracytoplasmic inclusions of proteins such as alpha-synuclein characterize these neurodegenerative changes that worsen over time. Lewy bodies, which are intracytoplasmic inclusions of the protein alpha-synuclein, and Lewy neurites are extracellular inclusions of the protein [68] are the neuropathological hallmarks of Parkinson’s disease.

Dopamine is a neurotransmitter that modulates a variety of brain functions and is involved in several pathways. The nigrostriatal pathway is a neuronal circuit that helps regulate movement [65]. Sixty percent to eighty percent of dopaminergic neurons are lost in PD, leading to motor symptoms. Resting tremors, bradykinesia, rigidity, or postural instability are all symptoms that help doctors make a diagnosis. Mood disorders such as anxiety and depression, sleep disturbances, fatigue, gastrointestinal disturbances, cognitive decline, and olfactory impairment are all examples of non-motor symptoms of Parkinson’s disease (PD). Parkinson’s disease can impact one’s QoL and ability to perform daily tasks in both motor and non-motor ways [69,70].

The beneficial effect of PBM (670 nm, 0.16 mW, 10 mW) on tyrosine hydroxylase-expressing neurons and Glial-Derived Neurotrophic Factors (GDNF) following 6-hydroxydopamine (6-OHDA) and MPTP (1-methyl-4-phenyl-1,2,3,6-tetrahydropyridine) lesioned striatal area in monkeys was evaluated in the study by El Massri et al. [71]. The study concluded that PBM demonstrated neuroprotective effects by restoring the loss of tyrosine hydroxylase-expressing neurons and increasing GDNF expression as compared to the control. In the same manner, it was reported that PBM (670 nm) treatment also influenced astrogliosis marker (GFAP) in MPTP (1-methyl-4-phenyl-1, 2, 3, 6-tetrahydropyridine) [72] and also synaptogenesis and brain-derived neurotrophic factor expression (BDNF) [73]. Despite the substantial evidence of the role PBM plays in the restoration of dopaminergic neuronal cell loss [71,72,74,75], little is known of its role in a lipopolysaccharide (LPS) model of dopaminergic cell loss; hence, the study by O’Brien and O’Brien and Austin [76] through the use of transcranial near-infrared photobiomodulation following supra nigral injections of 10 µg and 20 µg LPS in Sprague Dawley rats showed attenuation of inflammatory changes which was, however, dependent on the dose of the LPS administered. This study puts forward and has further substantiated the role of PBM in the restoration of dopaminergic neuronal cell loss and attenuation of inflammation which is an early event in PD and has further suggested the translation from bench to the bedside, the use of PBM as a veritable therapy in the management of neurodegenerative diseases.

Alim-Louis Benabid and his team in Grenoble have developed an implantable light-emitting device that has been tested in mouse, rat, and monkey models of PD by embedding the light-emitting device close to the midbrain. Intracranial PBM prevented the death of dopaminergic cells caused by 1-methyl-4-phenyl-1,2,3,6-tetrahydropyridine (MPTP) and 6-hydroxydopamine, and it decreased the level of severity of the resulting functional deficit, in animal models [77].

Remote photobiostimulation, a procedure that involves irradiation to a specific tissue and the surrounding non-irradiated tissue also being protected has also been established [78]. Following the discovery of remote photobiostimulation, Johnstone et al. [79] found that irradiating the dorsum of mice with 670 nm light and protecting the head with aluminum foil reduced substantia nigra dopaminergic neuronal cell loss in the MPTP mouse model of PD. This was further established in another study using a different mouse strain [80].

Recent transcriptomic results from Ganeshan et al. [81] lend credence to the idea that remote PBM activates signaling systems within the brain that recruit stem cells, bolstering the case for more targeted future studies to directly examine whether remote PBM stimulates the peripheral mobilization of stem cells and their recruitment to injured regions of the brain.

### 8.3. Traumatic Brain Injury (TBI)

Treatment options for traumatic brain injury (TBI) remain a challenge to date. Alterations in cerebral blood flow (CBF) are also linked to traumatic brain injury, with a decrease in blood flow following unresponsive vasodilation likely caused by nitric oxide release in the tissue [82]. Likewise, acute TBI models in several animal studies have been used to recapitulate the human form of TBI because of the impossibility of modeling chronic TBI in animals [83]. After recovering from a moderate or severe head injury, it is quite common for people to experience a wide range of persistent symptoms, some of which include depression, headaches, sleep disturbances, and cognitive impairment (such as poor memory, impaired executive function, and difficulties concentrating) [83].

The potential benefit of photobiomodulation on traumatic brain injury [84,85] was earlier documented in the literature. The study sought to examine the role of the application of red or near-infrared (NIR) on the scalp with the hope that it would be able to improve the cognitive deterioration experienced by mild and traumatic brain-injury patients. In their study, patients with chronic and mild traumatic brain injuries resulting from either motor vehicle accidents, home accidents, or sports injuries were recruited into the study following which a light emitting diode (LED) (5.35 cm diameter, 500 mW, 22.2 mW/cm^2^) was administered on the head for 10 min for 6 weeks (18 sessions; 3 times weekly). Before and after the LED administration period, several neurophysiological tests were conducted to ascertain the pre- and post-effect of the application of the LED treatments. It was concluded from the study that there were significant improvements in cognitive deterioration following treatments.

Because of the involvement of the mitochondrial complex IV electron transport chain enzyme (cytochrome C oxidase) [86,87] in the photobiomodulation process and also its role as a photoacceptor following biostimulation, with the knowledge that the mitochondria are compromised during traumatic brain injury [88,89], interventions towards the enhancement of mitochondrial functions are necessary. According to Naeser et al. [85], the putative mechanism by which photobiostimulation may lead to significant improvements in traumatic brain injury is via the increase in mitochondrial ATP production which could enhance cellular respiration, oxygenation, and function. It is known that in cells whose mitochondria are compromised, nitric oxide inhibits cytochrome c oxidase. Therefore in response to red/NIR photons, nitric oxide is released and diffuses outside the cell wall, leading to local vasodilation and increased blood flow [85]. Additionally, transcranial photobiomodulation demonstrated anti-inflammatory effects in different brain pathologies brought about by inflammation [90].

It is plausible that the mechanism by which PBM improves cerebral blood flow [91,92] and delivery of oxygen to the brain parenchyma is through enhancement of nitric oxide production [93,94].

### 8.4. Stroke

A stroke is classified as a cerebrovascular accident (CVA) and presents a significant socioeconomic burden on the sufferers of this condition. Naeser et al. [95] had earlier reported the significant benefit of transcranial photobiomodulation in stroke. In a recent case study by Stephan et al. [96], laser photobiomodulation was proven to show significant improvements in stroke and aphasia.

In rat stroke models caused by either permanent middle cerebral artery occlusion (MCAO) via craniotomy or by insertion of a filament, LLLT has been demonstrated to improve neurological deficits [97,98].

### 8.5. Epilepsy

Recurrent unprovoked epileptic seizures are the hallmark of epilepsy, which is produced by aberrant discharges of electrical activity in brain cells and can result in abnormal behavioral patterns [99,100,101]. Epilepsy affects both sexes equally at the rate of about 1% of the global population [102]. The majority of people with epilepsy have temporal lobe epilepsy (TLE) [103]. Studies are now beginning to explore the role of phototherapy in this condition. An evaluation of the effects of infrared lasers of varying powers on key neurotransmitters in the cortex and hippocampus (glutamate, aspartate, glycine, GABA, and taurine) uncovered that daily laser irradiation at 90 mW produced the most pronounced inhibitory effect in the cortex after 7 days [104]. Additionally, following pilocarpine administration in a rat model of epilepsy, it has been reported that laser therapy lowered the major neurotransmitters involved in epilepsy [105].

Excitotoxicity-mediated cellular death is an integral phenomenon in epilepsy. Low-level laser light therapy has proven to possess beneficial effects on primary neuronal cultures treated with excitatory amino acid neurotransmitters. Increases in ATP, mitochondrial membrane potential, intracellular calcium concentrations, oxidative stress, and nitric oxide levels were all significantly improved by low-level laser therapy (LLLT; 3 J/cm^2^ delivered at 25 mW/cm^2^ over 2 min) [106].

### 8.6. Major Depressive Disorders (MDD)

Major depressive disorder, also known as MDD, is widely regarded as being among the most urgent issues pertaining to mental health. Over the past 30 years, there has been an almost 50% increase in the number of incident cases across the globe, and there are currently more than 264 million people of all ages who are affected [107,108,109]. Notably, the present pharmacological interventions for depression present with side effects [110], likewise some types of depression are pharmaco-resistant [111]. It is therefore imperative to identify effective therapeutic interventions free from side effects, safe, economical, and non-invasive.

Several reports highlighting the role of photobiomodulation on animal models of depression have been documented. The reserpine model of depression was studied with the use of transcranial low laser irradiation by Mohammed [112]. In the study, transcranial laser irradiation of rats with different doses of (80, 200, and 400 mW), wavelength (804 nm) one week post reserpine (0.2 mg/kg) administration was sufficient to ameliorate the depressant effect following reserpine administration.

Furthermore, in a study by Xu et al. [113], LLLT was used to stimulate shaved scalp in a mice model of depression. The power output density and wavelength used in the study were 23 mW/cm^2^ and 808 nm, respectively, and laser irradiation was performed for a total of 28 days singly, for 30 min each day. LLLT at 808 nm resulted in reduced immobility and increased motor activity while at the same time mitigating depression-like behaviors. Several other studies have also reported the beneficial roles of photobiostimulation in depression [114,115,116,117].

### 8.7. Involvement of Remote (Systemic) Photobiostimulation

It is plausible that the delivery of light photons remotely could also provide some beneficial effects to the brain, perhaps by downregulating pro-inflammatory and upregulating anti-inflammatory cytokines [118,119]. Remote biostimulation (peripheral stimulation) may possess some beneficial effects for the central nervous system. The indirect application of near-infrared light in an MPTP-mouse model of Parkinson’s disease had some neuroprotective effects [79]. In their experiment, mice were treated with either direct NIR delivery to the head or the body. This study tested the hypothesis that NIR treatment of a remote tissue can induce systemic mechanisms that “indirectly” protect the brain. Few studies have examined such an indirect action, even though there is still accumulating evidence in this regard. These few studies have found remote, often bilateral, effects on tissues after local irradiation of skin wounds, gliomas (implanted on the dorsum of mice and irradiating abdomen), skin abrasions, and oral mucosa lesions [120,121,122,123]. Remote ischemic preconditioning also protects the brain, heart, and lungs from stress [124,125].

### 8.8. Effect of Polarization in Photobiomodulation

It is known that different properties or parameters may influence the effect of photobiomodulation in biological tissues, for example, polarization [126]. According to Mester et al. [4,5], parameters such as the fluence, the wavelength, the irradiance, the numbers and times of exposure, etc., may influence the effect that photobiomodulation possesses on tissues. More recently, Huang et al. [127] and Hadis et al. [128] also reported in their study on certain parameters (about 10) to be put into consideration for a much more effective photobiomodulatory action. These parameters include fluence, wavelength, irradiation time, energy, parameters of the pulses, beam area, power, number, the interval between the treatments, and the location of the tissue.

Polarization, which is the property that confers on light the ability to travel in a single plane is now the focus of recent studies [129]. The process of converting unpolarized light (light wave traveling in more than a single plane) to polarized (light wave traveling in a single plane) is known as polarization. Examples of unpolarized light sources include light from the sun, light from lamps, and tube light. In unpolarized light, there is a constancy in the direction of light propagation but with differing planes of change in amplitude. Light polarization is made possible through the use of a polarizer which blocks all other vibrations or propagation making light waves (photons) pass through a single plane. This may give a much more direct effect on biological tissues.

Photobiomodulation with polarized light was able to improve chronic lesions in patients [129]. The evidence that polarization of light may have a better advantage than non-polarization has been documented [126].

Linear, circular, and elliptical types of light polarization have been discussed depending on the orientation of the electric field [126,130]. The linear and circular types of light polarization have demonstrated significant effects at the cellular level when compared with unpolarized light. In the study by Ando et al. [131], they investigated the role of light polarization (800 nm LLLT) on spinal cord injury (SCI) in rat models. They found that treatment efficacy significantly improved with parallel polarization (based on the direction in which light was applied) as compared with perpendicular polarization.

In the study by Tada et al. [132], they sought to investigate the effect of light-emitting diode treatment on wound healing using polarized light on wound healing. They found that treatments with polarized light (linear and circular polarization) increased the wound healing process compared to unpolarized light. Hamblin [133] and Tripodi et al. [134] have also reported the role of polarization in photobiomodulation in a recently published work.

Based on the premise that there are currently few studies that have looked into the role of polarized and unpolarized light photobiomodulation effects and because the focus of photobiomodulation is in the use of red and near infrared light with no ablative effects in the treatments of various conditions, particularly the neurodegenerative diseases on which this review has focused, it is, therefore, necessary that researchers in the field of photobiomodulation consider polarization as an important parameter when utilizing this procedure.

### 8.9. PBM in Other Degenerative Diseases

Despite several reports on the beneficial role of photobiostimulation on certain neurodegenerative diseases, there is a paucity of information on the role of photobiostimulation on frontotemporal lobe degeneration, Amyotrophic Lateral Sclerosis, Huntington’s disease, and several other neurodegenerative diseases as depicted in Figure 3. Because of the socioeconomic burden and the reduction in the Quality of Life (QoL) of the sufferers of these conditions, more pre-clinical and clinical studies are needed to ascertain the role of PBM in their management, and likewise considering the different properties that may influence light stimulation of biological tissues.

## 9. Nanotechnology and Photobiomodulation Combination as Cutting-Edge Techniques in the Management of Neurodegenerative Diseases: A Consideration?

The production of materials with dimensions between 1 and 100 nm, which is a scale at which the properties of materials change dramatically from those of bulk materials and at which the properties can be tuned to the intended purpose, is the general focus of nanotechnology [135]. The use of nano-based pharmacotherapy in the management of neurodegenerative diseases (Alzheimer’s, Parkinson’s, schizophrenia, depression, and bipolar disorders) is beginning to gain recognition, as reported in several articles [136,137,138,139,140,141,142,143]. Different nano-based materials have been developed [144,145]. Gold [146,147], and carbon-based [148] and semiconductor-based [149,150] nanomaterials have shown significant results in neurodegenerative diseases [151,152]. These materials could form the composition of, or could be used themselves, in developing nanoparticles. These nano-based materials are photosensitive using the near-infrared (NIR) light spectrum [153]. The wavelength of NIR typically used in combination with nano-based materials for the delivery of drugs is about 700–1200 nm [154].

The fundamental features of a good nanodrug-delivery system include non-toxicity, innocuity, and biodegradability, but they also need to be able to avoid the immune system in vivo and be selectively carried across the blood–brain barrier (BBB) following delivery [153]. The blood–brain barrier (BBB) is a selectively permeable membrane that prevents the entry of harmful substances from systemic circulation into the brain. Despite the protective role of the BBB, it could also limit the delivery of certain therapeutic agents into the brain [155]. Some strategies earlier developed for the delivery of therapeutic agents that could bypass the BBB have presented structural integrity damage to the BBB [155]. To be able to overcome this challenge, researchers are now beginning to develop nano-based technology with no perceived effect on the BBB.

Nano-based drug delivery has been documented to possess significant advantages over the traditional mode of drug delivery, due to this, nanotherapeutic strategies have been proposed for the management of intractable brain diseases such as gliomas [156]. Due to their photothermal conversion effects and novel approaches to implementing on-demand drug delivery, photosensitive nanoreagents have garnered the most attention among numerous nanodrugs [156].

In light of this, because of the beneficial role of photobiomodulation using red and near-infrared light with a great tissue-penetrating ability, ablative effects, and the spatiotemporally controlled advantages light possesses [157], which is especially helpful for biomedical uses where precision and accuracy are crucial, studies are now beginning to examine the combination of nanotechnology with photobiomodulation.

### Strategies for the Nano-Based Photo (Light)-Controlled System

According to Bansal and Zhang [157], there are three main strategies developed for a nanoparticle-based light-controlled delivery system: (i) the use of light of a particular wavelength in activating “caged” bioactive molecules bound to photolabile groups and loading them onto a carrier; (ii) the use of nanocarriers that are photo-responsive; (iii) the use of nanoparticles made up of gold (Au), silver (Ag), sulfides, and graphene-based nanomaterials. Noble metal-based nanoparticles exhibit a particular property known as Surface Platon Resonance (SPR), which is a phenomenon of the resonance effect of the electrons within a nanoparticle following excitation by light photons (energy) [151]. Tuning the SPR wavelength from the visible to the NIR is possible by adjusting the size, shape, and composition of the nanoparticle size. Of all the three strategies, the use of nanoparticles made of Au has very good advantages in the field of nanomaterials because they can be easily synthesized, have good compatibility with biological tissues, and can be easily modified [151]. For a NIR drug-delivery system, Au seems to have a good ability to penetrate the BBB and possesses good photothermal properties making it an excellent choice in the field of nano research for the treatment of neurodegenerative diseases [140,147].

Meanwhile, several advantages could also be derived from using semiconductor polymer materials: (1) entirely made of organic material, fully non-toxic in living beings, and exhibiting excellent biocompatibility with living things; (2) with the capacity to adapt the synthesis and make the preparation simple, as well as having good photostability and outstanding photothermal properties [158,159,160]. It has been discovered that SPN when paired with NIR laser can modulate neuronal activity [153].

Perhaps the benefit of this nano-based light-controlled system includes brain-region specificity and accuracy in the treatment of different neurodegenerative diseases [153]. The application of nanotechnology in combination with photobiomodulation may present a significant breakthrough in overcoming some of the challenges seen in photobiomodulation and enable it to gain worldwide recognition in the management of several neurodegenerative diseases.

## 10. Clinical Trial Studies on Photobiomodulation

Notably, one of the challenges surrounding basic research is the translation from the bench-side to the clinic, because most basic research has been difficult to recapitulate in human subjects. Lately, several studies are beginning to examine the beneficial effect of photobiomodulation in humans. To this end, the study by Zhao et al. [161] sought to examine the role of photobiomodulation in patients with subjective cognitive decline (SCD). Subjective cognitive decline (SCD) has been defined by experts as the perception by self of cognitive decline [162,163] and this has been known to be a major risk factor for dementia brought about by Alzheimer’s disease. In the same manner, sleep has been known to significantly improve SCD. Because of the involvement of the frontal cortex in cognitive decline and disruption in sleep patterns, the authors aimed to use phototherapy to improve frontal cortex functions in SCD patients. In their study, fifty-eight subjects were recruited. Twenty-six patients received light therapy while thirty-two patients were the sham control, while the study period lasted for 6 days. Assessment of the sleep pattern from these study groups showed significant improvements in sleep quality and memory. This suggests that the role photobiomodulation plays in the improvement of cognition is not relegated to animal studies alone.

Despite several years of extensive research on neurodegenerative diseases, particularly Alzheimer’s disease, the available therapies are only symptomatic and may have some side effects. This may be because both animal and in vitro models have not been able to recapitulate essential aspects of the disease. Most drugs that have been discovered have likewise failed clinical trials. This calls for urgent attention to the need for non-pharmacological interventions with high efficacy and efficiency with no side effects such as photobiomodulation.

Berman et al. [164] conducted a double bind, placebo-controlled clinical trial on patients with mild cognitive impairment (MCI). The inclusion criteria for the study included the diagnosis of cognitive decline with a mini-mental status examination (MMSE) of 15–25 out of 30, and confirmation of dementia through the use of CT or MRI SCAN. In the study, near-infrared light (1072 nm) was applied to the frontal cortex for 6 min a day over 28 days in a bid to improve mitochondrial ATP and neural plasticity. Although this did not reach statistical significance, it should be noted that there was an improvement in executive functioning, EEG amplitude, as well as several other cognitive tasks.

Chao [165] also sought to examine the role of photobiomodulation in eight dementia patients subjected to PBM or the usual care for 12 weeks. In the study, the Alzheimer’s Disease Assessment Scale-cognitive (ADAS-cog) subscale and the Neuropsychiatric Inventory (NPI) were used to assess cognitive functions. NIR treatment was given to the patients (n = 4) at home while the usual care patients (n = 4) also received care. The baseline values obtained were compared with the post-treatment values. It was noted that the baseline values did not differ significantly between the two study groups. Interestingly, after 12 weeks of NIR treatments, it was noted that there were significant improvements in the ADAS-cog and NPI tests. The findings obtained from this study were also consistent with the study of Saltmarche et al. [166] following treatment with 810 nm NIR in mild and moderately cognitively impaired patients for 12 weeks using the ADAS-cog and mini-mental status examination (MMSE) tests.

Because of the differences in the brain anatomy of animals and humans and the use of photobiomodulation in enhancing different brain activities, caution must be taken however, when interpreting results obtained from animal studies to be able to model appropriately and also recapitulate photobiomodulation mechanisms for much better clinical efficacy.

## 11. Conclusions

Initially, the notion that light could have such far-reaching beneficial effects seemed implausible. However, the increasing amount of evidence supporting the concept suggests that it is pertinent to explore this technique to be able to harness its benefits in the management of several neurodegenerative conditions. Due to this increasing evidence on the use of photobiomodulation as a novel therapeutic target in the management of different neurodegenerative disorders, consideration should be given to the different optical properties and parameters which could influence this important photochemical application; additionally, there is a need for bed-side to bench-side translational research in addition to the already existing pieces of evidence to explore the role of PBM in several neurodegenerative diseases with the hope that it may serve as an alternative therapy in certain drug-refractory neurodegenerative diseases such as frontotemporal lobe dementia, amyotrophic lateral sclerosis, and Huntington’s diseases for which currently there is a paucity of information.

## 12. Future Perspective

Despite documentation on the beneficial effects of photobiomodulation in different animal models of neurodegenerative diseases (Alzheimer’s disease, Parkinson’s disease, and epilepsy) and other brain disorders, the clinical application of this procedure in animal studies of frontotemporal lobe degeneration, amyotrophic lateral sclerosis, and Huntington’s disease remains a virgin land that needs to be explored (Figure 3 and Figure 4). On the same note, the use of remote biostimulation (peripheral stimulation) which may possess some effects on the central nervous system could be an interesting area to explore.

The use of nanotechnology and light-controlled (photosensitive) nano-based material drug-delivery systems in the management of several neurodegenerative diseases may be a potential therapeutic target that needs exploration.

## Figures and Tables

**Figure 1 biomedicines-11-01828-f001:**
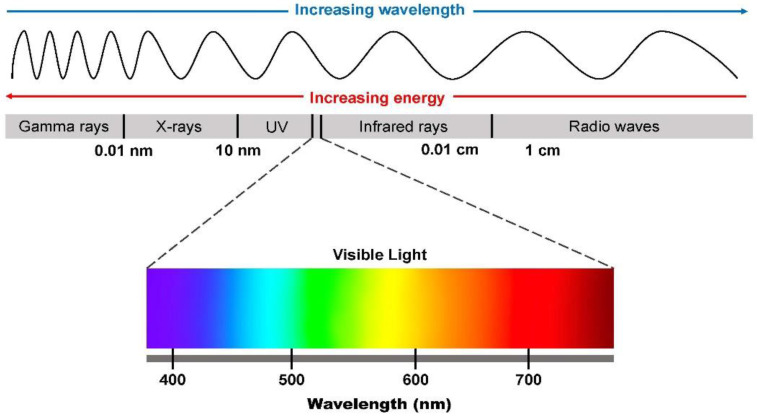
The electromagnetic spectrum (gamma rays, X-rays, ultraviolet rays, infrared rays, and radio waves) with two optical properties showing an inverse relationship (increasing wavelength with decreasing energy (photons) and vice-versa). The visible light spectrum between ultraviolet and infrared rays is magnified. Photobiomodulation utilizes wavelength within the red light to near infra red light spectrum for therapeutic purposes.

**Figure 2 biomedicines-11-01828-f002:**
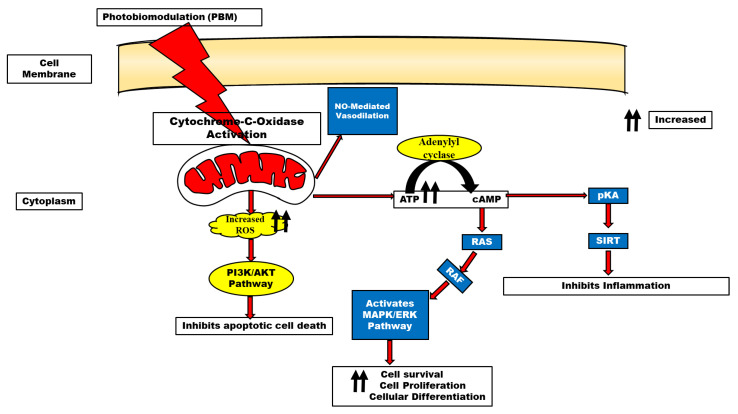
Schematic representation of photobiomodulatory effects through its action on the mitochondrial cytochrome C oxidase and activation of several signaling molecules. Following photobiostimulation, the mitochondrial chromophore, cytochrome C oxidase lead to ROS generation which in turn activates the PI3K/Akt pathway, while the effector enzyme, adenylyl cyclase, converts ATP to cyclic AMP (cAMP). cAMP could either activate pKA and RAS which further leads SIRT and ERK Signaling to promote cell survival, inhibit inflammatory processes, and apoptosis. NO release could also lead to NO-mediated vasodilation.

**Figure 3 biomedicines-11-01828-f003:**
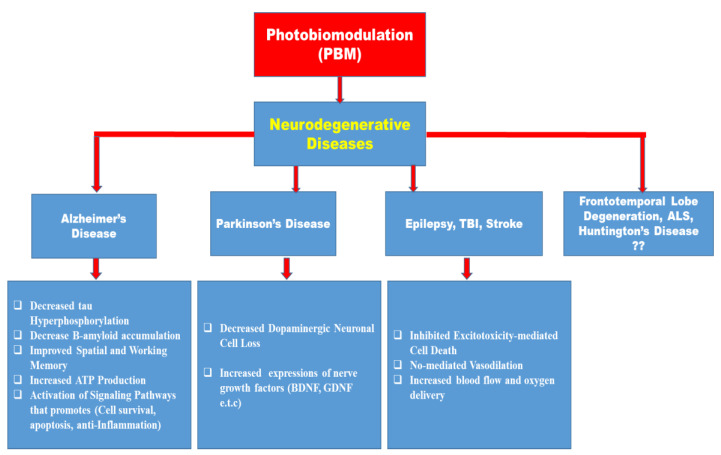
Schematic summary of the reported beneficial effects of photobiomodulation on neurodegenerative diseases (AD, PD, TBI, epilepsy) and the need to research other neurodegenerative diseases (frontotemporal lobe degeneration, ALS, and Huntington’s disease) for which there is presently a paucity of information.

**Figure 4 biomedicines-11-01828-f004:**
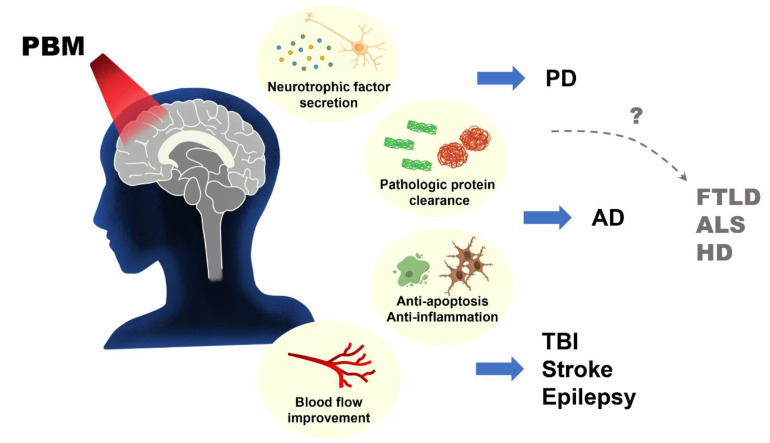
A graphical abstract showing the effects of photobiomodulation (PBM) in neurodegenerative diseases (neurotrophic factors secretion, pathologic protein clearance, anti-apoptotic, anti-inflammatory, and improvement in blood flow) and future directions on its application (FTLD, ALS, HD).

## Data Availability

This review article did not involve the collection of any new data.

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
