# Peer review of "The Beneficial Role of Photobiomodulation in Neurodegenerative Diseases"

_biomedicines, 2023, doi:10.3390/biomedicines11071828_

Round 1
Reviewer 1 Report
The manuscript ID biomedicines-2377322 has been devoted to mainly analyze the beneficial role of particular photobiomodulation effects with applications in neurodegenerative diseases therapies. Please see below some points to the authors of this review submitted to the prestigious journal Biomedicines:
1. My major concerns is that most of the subsections require better details and a deeper analysis to improve the discussion of this important topic.
2. The number of references for an original research paper is around 40; the number of references expected for a review is 200. The authors should take this consideration in the presentation of their work.
3. A graphical abstract showing the aim of the work and the considerations in the analysis at the beginning of the manuscript would be helpful for the audience.
4. A roadmap showing the evolution of photobiomodulation up to date would be welcome.
5. Photothermal effects induced in biological media have to be described by fractional modeling in advanced techniques. The authors are invited to discuss about this consideration as a perspective. You can see for instance: https://doi.org/10.1016/j.ijthermalsci.2022.107734
6. Please comment if polarization should be considered or it can be neglected to improve photobiomodulation in neurodegenerative diseases.
7. The authors should state what this review adds to literature in respect to others publication in the topic of photobiomodulation. You can see for instance: https://doi.org/10.3389/fnins.2022.1006031
8. Advantages and disadvantages of different wavelengths employed for photobiomodulation in neurodegenerative diseases should be mentioned.
9. The collective form of references should be presented in individual form (with individual expressions) in order to better justify their importance to be part of this review.
10. It is clear that the bibliography should be updated.
A proofreading is mandatory and a better use of language to connect the analysis and discussions would be welcome.
Author Response
Please see the attachment, thanks.

Reviewer 2 Report
Light is known to interact with some molecules to produce a detectable effect. Perhaps the most obvious of these involve vision, sunburn and vitamin D biosynthesis. There is a wide range of chromophores in living tissues that can interact with light. A collection of assorted mechanisms have been postulated to account for effects observed. Among the pertinent issues are the relative transparency of tissues to different wavelengths of light, the presence of appropriate chromophores, and reliability of studies claiming to demonstrate efficacy.
The important parameters for photophysics are wavelength and light dose. Transparency of tissues to light will vary with wavelength: only red and IR wavelengths will penetrate to any significant depth. But even here, penetration to depths of more than 1 cm are unusual. What is rare in an explanation of pertinent mechanisms. In order for light to interact with tissues, it must be absorbed, not scattered. There will need to be some biochemical interaction that produces an effect.
At this point, there appears to be some evidence for biochemical mechanisms associated with impingement of photons. There are many biologic effects that are clearly associated with light as a catalyst. Among these are Vitamin D biosynthesis, photosynthesis (in plants), sunburn, adverse effects associated with porphyria. Mechanisms for all of these phenomena have been thoroughly described. Effects on neurodegenerative diseases may be documented, but mechanisms are only dimly perceived. This report does discuss some of the possibilities.
The photochemical aspects of photobiomodulation appear to be correctly discussed. It is clear that penetrating wavelengths of light are needed. Chromophores continue to be identified. Whether there is a significant role for this approach in clinical medicine will need to be reviewed by neurologists and other medical personnel.
Author Response
Please see the attachment, thanks.

Reviewer 3 Report
The manuscript by Ayodeji Zabdiel Abijo et al discusses the role of Photobiomodulation in neurodegenerative diseases. Although the topic might be of interest as non pharmacological therapy it looks not original in its form since it recapitulates the information already reported in the literature without any additional new details. It would be of interest to refer on the clinical application of the photobiomodulation in neurodegenerative diseases in order to strength the relevance of the new alternative therapeutic approach.
The manuscript is not original in its form and does not add any new information on the topic of Photobiomodulation application for the treatment of neurodegenerative diseases, compared to the other reviewes present in the literature on the same topic.
The authors might discuss about the clinical trial on photobiomodulation in neurodegenerative diseases in order to make the manuscript more appealing on the scientific point of view.
Author Response
Please see the attachment, thanks.

Round 2
Reviewer 1 Report
The authors of the manuscript ID biomedicines-2377322 have partially improved the presentation of their work by including a study related to the effect of polarization in photobiomodulation and clinical trial studies. However, important issues previously raised have been omitted to enhance the discussion by considering cutting-edge techniques and other essential considerations involved in photobiomodulation. Better details and a deeper analysis is still missing in order see what this work adds to literature beyond what has been already said in the papers selected to be mentioned in a very general form within this review.
A better use of language to connect the analysis and discussions would be welcome.
Author Response
Please see the attachment, thanks.

Reviewer 3 Report
Although the effects of the authors to improve the quality of the review accomplishing the referee's suggestions, the study still lack of novelty.
Author Response
Please see the attachment, thanks.

Round 3
Reviewer 1 Report
As I see, Reviewer #1 has recommended an additional revision of the paper while the Reviewers #2 and #3 agree that better details and a deeper analysis is still missing in order see what this work adds to literature beyond what has been already said in the papers selected to be mentioned in a very general form within this review.
At this point, the revision of the manuscript is still unsatisfactory. No additional analysis suggested has been incorporated to highlight the originality that is standard in a review. Cutting-edge techniques and other essential considerations involved in photobiomodulation recommended to be reviewed have been omitted.
A proofreading is mandatory.
Author Response
Please see the attachment, thanks.

Round 4
Reviewer 1 Report
The authors have included an additional section that improved the work and I agree with the focus of the topic. Then I consider that the text worth publication in present form.
A standard proofreading would improve the quality of English